# Gut Microbiota Composition in Undernourished Children Associated with Diet and Sociodemographic Factors: A Case–Control Study in Indonesia

**DOI:** 10.3390/microorganisms10091748

**Published:** 2022-08-30

**Authors:** Mifta Gatya, Dwi Larasatie Nur Fibri, Tyas Utami, Dian Anggraini Suroto, Endang Sutriswati Rahayu

**Affiliations:** 1Faculty of Agricultural Technology, Universitas Gadjah Mada, Yogyakarta 55281, Indonesia; 2Center of Excellence for Probiotics, Universitas Gadjah Mada, Yogyakarta 55281, Indonesia; 3Center for Food and Nutrition Studies, Universitas Gadjah Mada, Yogyakarta 55281, Indonesia

**Keywords:** undernourished, diet, sociodemographic, gut microbiota, dysbiosis

## Abstract

Malnutrition, which consists of undernutrition and overnutrition, is associated with gut microbiota composition, diet, and sociodemographic factors. Undernutrition is a nutrient deficiency that that should be identified to prevent other diseases. In this study, we evaluate the gut microbiota composition in undernourished children in association with diet and sociodemographic factors. We observed normal children (n= 20) and undernourished children (n= 20) for ten days in Lombok and Yogyakarta. Diet, sociodemographic factors, and medical records were recorded using food records, screening forms, and standard household questionnaires. Gut microbiota analysis was performed using 16S rRNA gene sequencing targeting the V3–V4 region. The result showed that the undernourished group had lower energy intake. In addition, the undernourished group had lower quality of medical records, parent knowledge, education, and exclusive breastfeeding. Firmicutes, Bacteroidetes, Actinobacteria, Proteobacteria, and Verrucomicrobia were significantly different between normal and undernourished children. Based on LefSe, we determined that *Akkermansia* is a biomarker for undernourished children. In conclusion, diet and sociodemographic factors affect the gut microbiota composition of undernourished children.

## 1. Introduction

Malnutrition is an imbalance of nutrient uptake or intake in the body [1] that can be represented either by overnutrition or undernutrition. Undernutrition is a nutrient deficiency that causes a change in body cell mass, resulting in loss of physical and mental function [2]. According to the anthropometric index, undernutrition consists of stunting (height/age < −2 standard deviation) and wasting (weight/age < −2 standard deviation) [3]. The WHO estimates that 149.2 million children are stunted and 13.6 million are wasted worldwide [4]. Various factors, such as diet and sociodemographic conditions, including age, gender, education, health status, and income, can cause differences in nutritional status. Research conducted in Indonesia revealed that 30.8% of children have experienced some form of stunting, with the second highest incidence of stunting cases in the province of Nusa Tenggara Barat (NTB) [5]. According to NTB health profile data, 29.5% of stunted children in the province reside in East Lombok Regency.

According to data from the Central Statistics Agency, Yogyakarta (15.1%) has the second lowest percentage of stunted toddlers in Indonesia [6]. Poor nutrition prevents children from reaching their full physical and cognitive potential, which causes them to live in poverty and have a low quality of human resources [7]. Therefore, it is necessary to practice early prevention with respect to the problem of undernourished children.

Undernourished children tend to undergo a condition called dysbiosis [8]. Dysbiosis is characterized by alteration of the gut microbiota composition, causing a decrease in immune system activity associated with the gut microbiota and the emergence of enteropathogenic bacteria [9]. An increased number of pathogenic bacteria can cause a decrease in the ability to absorb nutrients and immune system activity, which leads to digestive tract inflammation and intestinal mucosa [10]. In undernourished children, an increase in *Enterobacteriaceae*, which belong to the Proteobacteria phylum, is associated with impaired absorption and intestinal protection against pathogens [11,12]. Some hypotheses suggest a correlation between gut microbiota composition and undernourished children as follows: undernutrition causes metabolic dysfunction, which is characterized by differences in the design of the gut microbiota; decreased immune system activity is related to the activity of the gut microbiota, and undernutrition is caused by a correlation between factors, such as impaired absorption of nutrients, which is associated with the emergence of enteropathogenic bacteria [13]. Some previous studies have discussed the gut microbiota composition of malnourished children without considering influencing factors. Thus, the aim of the present study is to determine markers and explore the gut microbiota profile of children with poor nutritional status in association with dietary factors and sociodemographic conditions.

Several studies have highlighted the relationship between nutritional status and gut microbiota composition. The gut microbiota comprises microorganisms that live in the digestive tract, the numbers of which are estimated to exceed 10^14^, which is ten times more than the number of human cells in the body [10]. Gut microbiota composition in the digestive tract increases immune system activity, accelerating metabolism and helping to protect the digestive tract from pathogens [14]. The gut microbiota composition is affected by several factors, including diet, lifestyle, environment, illness, and medication habits [15]. Generally, children in Yogyakarta have the *Prevotella* enterotype, which is associated with a high-carbohydrate diet.

Individuals who consumes a high-protein diet tend to have a *Bacteroides* enterotype [16]. Place of residence is another factor that affects gut microbiota composition. According to De Filippis [17], *Succinivibrio* can be used as a marker of subjects who live in rural areas. *Succinivibrio* is associated with degradation of starch, hemicellulose, and xylan. It is not found in subjects who live in urban areas. 

We conducted a pilot-scale case–control study of undernourished children in East Lombok and normal children in Yogyakarta to explore the effect of diet and sociodemographic factors on gut microbiota composition in undernourished and normal children. Such research needs to be performed in an early stage to understand the characteristics of the gut microbiota in undernourished children. The results of this study can be used as a reference for parents to prevent malnutrition in children through proper dietary interventions that could modulate the gut microbiota composition.

## 2. Materials and Methods

### 2.1. Ethical Approval and Protocol Registry

The study protocol was approved by the Medical and Health Research Ethics Committee (MHREC), Faculty of Medicine, Gadjah Mada University (approval reference: KE/FK/0502/EC/2022), on 25 April 2022. This research was conducted in accordance with the applicable Indonesian regulations and guidelines of the World Medical Assembly (Declaration of Helsinki) with notes of CPMP on GCP (CPMP/ICH/135/95). Informed consent was signed by the parents/legal guardians of participants prior to the observation.

### 2.2. Study Design

Sixty children aged 8–12 years were recruited for this research through screening based on inclusion and exclusion criteria. These 60 subjects were classified into two groups related undernutrition and normal status. Children who were included in the undernourished group were domiciled in Belanting, Lombok Timur, with a body mass index < 13.6 kg/m^2^ (female) or <13.9 kg/m^2^ (male). The normal group was domiciled in Yogyakarta, with a body mass index in the range of 13.6–18.4 kg/m^2^ (female) or 13.9–18.0 kg/m^2^ (male) [18,19]. The aim of this research was to evaluate the dietary patterns and sociodemographic factors associated with the gut microbiota composition of elementary students from Lombok and Yogyakarta. Qualified subjects were observed for ten days and filled out a questionnaire, providing information on demographic data, the subject’s daily activities, medical records, and dietary records. On the 11th day, subjects provided a fecal sample (using sampling kits provided by the researcher) and submitted their questionnaires. The height of children participating in the study was measured with a microtoise (0.1 cm accuracy), and a digital scale was used to measure weight. Sixty children were identified as eligible to join the research through the screening procedure, but only forty fulfilled data requirements. 

### 2.3. Dietary Information

The subject’s dietary pattern was collected by recording their daily meals, ingredients, and quantity for ten days. The energy and the amount of each nutrient were estimated according to dietary records according to the 2007 NutriSurvey supplemented of the Indonesian food database [20].

### 2.4. Fecal Sample Collection

Fecal samples were collected on the 11th day in sterile tubes containing 2 mL RNA (Sigma Aldrich; R0901; Saint Louis, MO, USA) as DNA/RNA stabilizers. Subsequently, the fecal samples were transported to the laboratory in a cooler box (<10 °C) within one hour of collection. Fecal samples were labeled and stored in a freezer at a temperature of −25 °C for further molecular analysis.

### 2.5. DNA Extraction

DNA was extracted according to protocol described by Nakayama, with some modifications [21]. First, samples were prepared by RNA-later 10-fold dilution, followed by washing with 1 mL PBS twice. The samples were then mixed with 300 μL of Tris-SDS solution and resuspended; 500 μL of TE buffer-saturated phenol was added to each sample. This mixture was then subjected to a bead beater (BeadBug Microtube Homogenizer, Benchmark Scientific; Shanghai, China) at a speed of 4000 rpm for 60 s. The obtained supernatant was combined with 200 μL of phenol/chloroform/isoamyl alcohol (25:24:1; *v*/*v*) (Sigma-Aldrich; P2069; Saint Louis, MO, USA) and resubjected to a bead beater at 4000 rpm for 90 s, followed by centrifugation at 13,000 rpm for 5 min at 4 °C (DLAB Scientific Co., Ltd.; Beijing, China). After centrifugation, the supernatant was mixed with 25 μL of 3 M sodium acetate (pH 5.2) (Sigma-Aldrich; 567422; Saint Louis, MO, USA), followed by the addition of 300 μL isopropanol, incubated for 30 min at −30 °C, and centrifuged at 13,000 rpm for 5 min at 4 °C. The DNA pellet was washed with 500 μL of 70% ethanol (minimum temperature of 10 °C) and centrifuged at 13,000 rpm for 5 min at 4 °C. The DNA pellet was air-dried at room temperature before being suspended in 20 μL of TE buffer (pH 8.0) and stored at −30 °C until use. 

### 2.6. 16S rRNA Gene Amplicon Sequencing 

16S rRNA sequencing analysis was performed at the Department of Bioscience and Biotechnology, Faculty of Agriculture, Kyushu University, according to the protocol described by Nakayama [20]. Stool genomic DNA was amplified using TaKaRa ExTaq HS (Takara Bio, Shiga, Japan), targeting the V3-V4 region (F (Bakt_341F): CGCTCTTCCGATCTCTGCCTACGGGNGGG- WGCAG, R (Bakt_805R): TGCTCTTCCGATCTGACGACTACHVGGGTATCTAATCC). The obtained amplicons were then used as a template for secondary PCR, attached with barcode-tag primers, mixed, and subjected to paired-end sequencing using Illumina sequencing (Illumina Inc., San Diego, CA, USA). 

### 2.7. Sequencing Data Processing 

The sequence data were processed using Usearch (v.9.2.64) to construct operational taxonomic units (OTUs) with 97% identity and to remove PCR chimeras [21,22]. The taxonomy of each OTU was analyzed using the SINTAX command in the Greengene (v.13.5) database (gg_16s_13.5.fa.gz) and a cutoff value of 0.8 [23]. Downstream analysis performed with Quantitative Insights into Microbial Ecology (QIIME) virtual-box pipeline software (v.1.9.1). In QIIME software, summarize_taxa_througy_plots.py and alpha_rarefaction.py commands were used to assign a taxonomic composition for each sample and calculate the alpha diversity index, respectively. Linear discriminant analysis effect size (LEfSe) was also applied using an online version of Galaxy to visualize significantly overexpressed specific microbes in each group as a biomarker [24]. The linear discriminant analysis (LDA) was performed using one-against-all criteria. The LDA score threshold was 2, and the alpha value was 0.1 for Kruskal–Wallis and pairwise Wilcoxon tests, respectively. 

### 2.8. Statistical Analysis 

Statistical analysis was conducted and graphics were generated using RStudio (v.4.0.3) and R studio (v.1.1.463), ggplot2 (v. 3.3.2), and vegan (v.2.5-6) packages (GNU Affero General Public License, San Francisco, CA, USA). Demographic and baseline characteristics were calculated using descriptive statistics for the total sample population and each group. In addition, Pearson chi-square analysis for categorical data and independent *t*-tests or Mann–Whitney tests for numerical data were used to assess the homogeneity of baseline characteristics between groups, depending on the fulfillment of the normal distribution assumption. Independent *t*-tests and Wilcoxon tests were used to analyze the experimental parameters during the intervention depending on the distribution assumption. The Wilcoxon rank sum test was used for comparison between groups, whereas permutational multivariate analysis of variance (PERMANOVA) was used to evaluate the gut microbiota composition at the OTU and genus level. In addition, non-metric multidimensional scaling (NMDS) based on Bray–Curtis dissimilarity was also used to visualize the difference between groups.

## 3. Results

### 3.1. Subject Characteristics

This research was a case–control study, without intervention. Figure 1 shows the CONSORT flow chart of the study procedure.

Forty participants were divided into two groups using characteristic criteria of body mass index and city to classify participants as normal or undernourished children. Subject characteristics are shown in Table 1.

Forty children aged 8–12 years were divided into two groups (normal nutritional status and undernourished status) based on their BMI. Children with normal nutritional status had a BMI of 15.98 ± 0.73, whereas undernourished children had a BMI of 13.53 ± 0.51. Body weight and BMI differed significantly between groups because as a result of the inclusion criteria specified in this research. 

### 3.2. Dietary Intake Differs between Normal and Undernourished Children

Dietary intake between groups was evaluated according to food record forms filled out by subjects for ten days (Table 2), showing that overall nutrient intake among children with normal nutritional status is higher than that of undernourished children, especially with respect to macronutrients and minerals.

All parameters show a significant difference, except for vitamin C intake (Table 2). Macronutrient, micronutrient, and mineral intake is below the recommended dietary allowance (RDA) for children aged 8–12 years in both groups. Only vitamins A, B6, and zinc intake in children with normal nutrition status meet the RDA standard. In undernourished children, intake is far below the RDA standard with respect to all nutritional parameters. Kemenkes et al. [5] stated that energy intake below 70% is associated with a lack of energy. In the present study, the normal and undernourished groups have energy intake of >80% and <40%, respectively. The result shows that undernourished group experiences a lack of energy intake.

### 3.3. Gut Microbiota Composition Differs between Normal and Undernourished Children

The gut microbiota composition of 20 undernourished children in East Lombok was compared with that of 20 normal children in Yogyakarta. 

As observed in the research, the major phyla in both groups were Firmicutes and Bacteroidetes (Table 3). In the normal group, the next most prominent group was Actinobacteria, whereas the presence of Verrucomicrobia was more pronounces than that of Actinobacteria in the undernourished group. Proteobacteria and Fusobacteria were present at concentrations of 1.96–2.56% and 0.1%–0.12%, respectively. Both Actinobacteria and Proteobacteria contents differed significantly between groups. Only Bacteroidetes and Fusobacteria did not show a significant difference. The ratio between Firmicutes and Bacteroidetes (F/B) can describe a condition called dysbiosis, which is an unbalance of gut microbiota composition [22]. The undernourished group had a higher ratio of F/B than the normal group (Figure 2). 

Gut microbiota composition at the genus level was selected based on the top 14 most abundant the genera (Table 4), including seven genera belonging to Firmicutes, two genera each belonging to Bacteroidetes and Actinobacteria, and one genus each belonging to Proteobacteria, Verrucomicrobia, and Fusobacteria. Some genera showed a significant difference in RA, such as *Ruminococcus*, *Bacteroides*, *Bifidobacterium*, *Collinsella*, *Succinivibrio*, and *Akkermansia*. The most abundant microbiota at the genus level was *Prevotella*, which was present at a concentration of 13.45–15.36% in both groups, whereas Fusobacterium was the least abundant.

The microbiota in the undernourished and normal groups was defined using the 16S rRNA gene. A total of 1,670,940 reads were obtained from 40 samples, resulting in 729 OTUs, with 15 phyla and 113 genera detected. Four metrics of alpha diversity were observed in this research: Chao1, observed OTUs, whole-tree PD, and Shannon index (Figure 3). According to the results, alpha diversity at baseline groups did not differ significantly between the two groups at baseline, including community evenness for species of OTUs, comparison of the phylogenetic tree, and richness between groups. Species abundance and diversity were significantly different between the normal and undernourished groups. Furthermore, undernourished children had a lower diversity of bacteria compared with the normal group. 

Beta diversity reflects the gut microbiota variation between groups. A significant difference was observed using the Bray–Curtis method (*p* = 0.001; R^2^ = 0.1643). Non-metric multi-dimensional scaling (NMDS) based on Bray–Curtis dissimilarity was used to visualize the difference between groups (Figure 4). The distance between data points reflects the similarities between groups. A zero value of the stress factor (R^2^) indicates considerable similarity, whereas a stress factor closer to 1 indicates poor representation. According to the results, the stress factor between groups is less than 0.2, indicating considerable dissimilarities in microbiota composition. Therefore, the two groups have different gut microbiota compositions, but the overlapped point in weighted and unweighted UniFrac indicates similar bacteria between the two groups.

According the result, two phyla and genera were significantly different in the normal group, whereas three phyla and genera were significantly different in the undernourished group. As shown in Figure 5, *Akkermansia*, which belongs to Verrucomicrobia, showed an LDA score of more than 10^4^. A cladogram was also generated from LefSe, showing the most abundant taxa of gut microbiota composition. Taxonomic levels are represented by rings; the diameter of each ring is proportional to its abundance. The cladogram visualized LDA results, with *Akkermansia* identified as a marker of the undernourished group and while Bacteroides as marker of the normal group.

### 3.4. Diet Affects Gut Microbiota Composition in Undernourished Children

The correlation determined using corrplot shows that the gut microbiota is affected by diet, especially by intake of macronutrients, such as protein, fat, carbohydrates, and dietary fiber (Figure 6). Major bacterial phyla in the gastrointestinal tract include Firmicutes and Bacteroidetes. Firmicutes are positively correlated with all micronutrients, the most significant being their correlation with carbohydrates. Firmicutes have a high capacity to ferment and metabolize carbohydrates, as well as lipids, which contribute to the development of obesity [23]. Firmicutes bacteria, such as *Ruminococcus*, produce enzymes responsible for carbohydrate and dietary fiber degradation [24].

Bacteroidetes is negatively correlated with all macronutrients, as well as BMI, showing that the relative abundance of Bacteroidetes is reduced with increased BMI. Furthermore, *Prevotella* and *Bacteroides* represent a gut microbiota enterotype influenced by diet. Research indicates that a high relative abundance of *Prevotella* is linked to plant-based dietary habits, whereas *Bacteroides* are linked a diet rich in animal protein [25]. The results show that *Bacteroides* are positively associated with all the macronutrients, whereas there is no significant correlation between *Prevotella* and macronutrients.

### 3.5. Sociodemographic Factors Affect Gut Microbiota Composition in Undernourished Children

Sociodemographic factors observed in this study were gender, age, environment sanitation, medical record (including delivery procedure and congenital disease), and exclusive breastfeeding (Table 5). Subjects who participated in this study were children; therefore, factors such as parents’ education, income, occupation, and knowledge were also included. In addition, BMI was included to represent the ratio of body weight to height in the normal and undernourished groups. These factors were then compared with gut microbiota composition. 

Body mass index, which represents nutritional status, is significantly positively correlated with parents’ education, income, medical record, sanitation, occupation, knowledge, and exclusive breastfeeding. The present study involved two groups, which explains why subjects with higher BMI also have higher values with respect to parents’ education, medical record, knowledge, and exclusive breastfeeding. With respect to relative abundance of the gut microbiota, we did not obtain a significant result relative to BMI. However, BMI is positively correlated with Actinobacteria, Firmicutes, and *Bacteroides* but negatively correlated with Bacteroidetes, Proteobacteria, and *Prevotella*. 

## 4. Discussion

Based on BMI parameters, children’s health status can be classified as undernourished/underweight, healthy, overweight, and obese. The undernourished group is characterized by a BMI lower than the fifth percentile, which is <13.7 [18,19]. One of the factors that cause undernutrition is insufficient dietary intake. In this research, the nutrition intake of the undernourished was lower than the RDA in all categories. Therefore, energy intake did not meet the standard in this group because it was below 70% of the RDA. All nutrient parameters differed significantly between normal group and the undernourished, except for vitamin C intake. Based on food records, the subjects in the undernourished group still consumed some fruits but preferred only to eat the same kind of fruit, such as papaya or banana (Appendix A). Most of the people in Belanting cultivate banana and papaya, making them common fruits in the diets of children residing in Belanting. The varieties of banana and papaya cultivated in Belanting are *pisang kepok* (*Musa acuminata balbisiana Colla*) and California papaya (*Carica papaya* L.), respectively. *Pisang Kepok* is high in vitamin C compared with other banana varieties [26,27], as is the case of papaya. Vitamin C accounts for the highest micronutrient content in papaya compared with other micronutrients [28]. The vitamin C content in these fruits could explain why vitamin C levels were similar between the undernourished group and the normal group, whereas the contents of other nutrients were significantly lower in the undernourished group.

Based on sequencing using 16S rRNA, each group had significant gut microbiota abundance and diversity results. The two groups showed a significant dissimilarity in gut microbiota diversity. The undernourished group had a lower relative abundance of gut microbiota with higher Proteobacteria content. These results are in accordance with research involving healthy and severe acute malnutrition (SAM) in Bangladesh, where it was found that in SAM children, proteobacteria accounted for 46% of the gut microbiota, much higher than Bacteroidetes, which accounted for only 18%. On the contrary, in healthy children, the abundance of Bacteroidetes was 44%, whereas that of Proteobacteria was 5% [29]. Phylum Proteobacteria includes several pathogens (*Salmonella enterica*, *Preudomonas xanthomarina*, *Hafania alvei*, *Klebsiella*, and *Escherichia coli*) [30]. Besides having a higher abundance of Proteobacteria, undernourished children also had a higher ratio of Firmicutes in comparison with Bacteroidetes (F/B). These results are in line with those reported by Mendez-Salazar et al. [31], who found that the malnourished group had a more higher F/B ratio than the normal group. Ratios among those phyla have been associated with homeostasis, and changes in this ratio can lead to various diseases or pathologies, such as obesity and bowel inflammation [32,33]. Firmicutes include Gram-positive bacteria, which can efficiently ferment and metabolize carbohydrates and lipids. In increase in this phylum can contribute to the development of obesity. Bacteroidetes include Gram-negative bacteria, which also comprise a variety of carbohydrate-degrading genes [23]. Previous research revealed that Bacteroidetes are often increased in inflammatory bowel disease (IBD) and are associated with its progression and development. However, Bacteroidetes content was also lower in the undernourished group relative to the normal group and contributed to the N-glycan pathway. This deficiency may result in less efficient energy extraction from non-digestible polysaccharides or diet fiber [34]. There was also a lower abundance of *Ruminococcus* and a higher abundance *Succinivibrio*, which belong to Firmicutes and Proteobacteria, respectively, in undernourished children relative to normal children. *Ruminococcus* increases carbohydrate intake and can degrade starch into simple sugar, allowing substrate fermentation to produce energy [35]. Besides being a marker for gut dysbiosis, *Succinivibrio*, which belongs to the Proteobacteria phylum and produces proinflammatory succinate, can also increase gut permeability [36,37].

The 40 children who participated in the present study can be grouped into four regions based on the PCA plot (Figure 7). Overall, *Prevotella*, which belongs to the Bacteroidetes phylum, dominated most groups. These results indicate that both groups had the *Prevotella* enterotype. This result is in accordance with the results reported by Nakayama et al. [16], who found that school-aged children in Asia, especially in Indonesia and Thailand, are mainly driven by the *Prevotella enterotype* (*p*-type). This enterotype reflects the a diet rich in carbohydrates and resistant starch. Meanwhile, subjects from China, Japan, and Taiwan were found to harbor the *Bacteroides* enterotype. This genus is correlated with high-fat diets and high levels of animal protein, which often induce obesity. Children who have a high BMI and calorie intake tend to have a high abundance of *Bacteroides*, caused by a high intake of animal protein. As shown in the pie chart in Figure 7, the gut microbiota composition between SUC and HC is similar and only differs in terms of abundance, although healthy children have a higher abundance of genera. This could be due to the fact that only a few dominant genera are plotted in the pie chart, and not all genera present in the pie chart.

As shown in Figure 7, the UC group is dominated by *Akkermansia*, which belongs to the Verrucomicrobiota phylum. *Akkermansia* was also found to be a marker for undernutrition in children based on LefSE and LDA score (Figure 5). This result contradicts the literature, which suggests that *Akkermansia* is a marker for digestive health status because of its ability to colonize the mucosal layer and produce acetate and propionate [38]. Its main species, *Akkermansia muciniphila*, is also inversely correlated with metabolic syndrome [39]. Previous studies have also proven that treatment with *A. muciniphila* can reduce the risk of obesity and improve insulin resistance and glucose intolerance [40,41]. Surprisingly, the results of the present study suggest that *Akkermansia* is a microbial marker for poor glycemic control and is enriched in subjects without continuous subcutaneous insulin infusion (CSII) therapy and non-controlled HbA1c level [42]. Diet and environment can account for the controversial findings related to *A. muciniphila.* Diet could modulate *Akkermansia* levels, as has been demonstrated in animal models undergoing a drastically reduced abundance of *Akkermansia*, which is restored after supplementation with *A. muciniphila* [43]. According to Ibragimova et al. [44], *A. muciniphila* is bacterial signature of plant-based diets, including vegan, vegetarian, and Mediterranean diets consisting of fruits and vegetables, whole grains, legumes, beans, fish, nuts, and seeds. A calorie-restricted diet, a reduced-energy diet, low fermentable oligo-, di-, monosaccharides, and polyols (FODMAP) also increased the abundance of *A. muciniphila* [45], which is in accordance with our finding that undernourished children primarily consume a plant-based diet, as evidenced by the dominance of the *Prevotella* enterotype, i.e., the second most dominant genus in undernourished children (Figure 7).

Moreover, the dietary intake of undernourished children, including macronutrient and micronutrient intake, is lower than the RDA standard. This could account for the increased abundance of *Akkermansia* in undernourished children, although there were no specific species of *Akkermansia* detected in the present study. However, *A. muciniphila* is the dominant species of *Akkermansia*.

In the present study, all sociodemographic factors (parents’ education, income, occupation, knowledge, children’s age, medical record, sanitation, and exclusive breeding), except medical record, were positively correlated with BMI (Table 5. A significant correlation was observed with medical record, exclusive breastfeeding, parents’ education, and knowledge, suggesting that children with higher BMI have parents with better education and knowledge concerning health and nutrition. Children who received exclusive breastfeeding also tend to have higher BMI. With respect to medical record, children with higher BMI rarely get sick. Based on the questionnaire results, common diseases among children include flu, diarrhea, cough, and fever. These findings are in accordance with previous research, which found that children who live in rural areas with lower socioeconomic class and whose mothers have anemia, poor literacy, low education, little knowledge about health and sanitation, and history of an excessive number of births are more likely to suffer a growth disorder, mainly undernutrition [46,47,48]. Some gut aspects of microbiota composition were also significantly correlated with sociodemographic factors: (1) Bacteroidetes were positively correlated with sanitation, parents’ education, and knowledge; (2) Firmicutes were positively correlated with age and medical record; (3) Proteobacteria were negatively correlated with income and parents’ knowledge; (4) *Prevotella* was positively correlated with sanitation and parents’ knowledge; and (5) *Bacteroides* were negatively correlated with age and exclusive breastfeeding. According to Nagpal et al. [49], there is a correlation between gut microbiota and aging. In infants, the gut microbiota is dominated by *Bacteroides* and *Bifidobacterium*, but *Bifidobacterium* is later be replaced with Firmicutes.

Furthermore, the enterotype adjusts to the dietary pattern, and the presence of *Bacteroides* is reduced with age. The decrease in the number of *Bacteroides* could be caused by the dietary habits of children who tend to consume a plant-based diet, as proven by the increasing abundance of *Prevotella*, which positively correlates with age. Proteobacteria, as a marker of undernourished status, is reduced in correlation with increased parents’ income and knowledge. As previously explained, Proteobacteria mainly consist of the pathogens, with lower abundance in contexts of better hygiene and sanitation. 

## 5. Conclusions

In the present study, undernourished children were found to lower gut microbiota diversity and a higher F/B ratio than the normal group, which indicates dysbiosis. Gut microbiota composition differed between the two groups, but there were some bacteria that were similar between groups. Compared with the normal group, undernourished children had lower Firmicutes, Bacteroidetes, and Actinobacteria and higher Proteobacteria and Verrucomicrobia contents. Genera *Ruminococcus*, *Bifidobacterium*, and *Collinsela* were present in lower concentrations, with increased abundance of *Succinivibrio* and *Akkermansia* in undernourished children relative to their normal counterparts. The gut microbiota of undernourished children were dominated by the *Prevotella* enterotype, which is associated with a plant-based diet.

Moreover, some sociodemographic factors, such as parents’ education and knowledge concerning nutrition, tend to result in a higher BMI, as well as children who were exclusively breastfed. *Akkermansia* is a biomarker for undernourished children, which is correlated with a calorie-restricted diet, reduced-energy diets, and diets with low FODMAP. The results of this research are expected to help prevent undernutrition from becoming a frequent problem in children. It is important to further knowledge regarding nutrition, not only for children but also their parents, as the condition of children is affected by their parents, especially their mothers. The knowledge generated in the present study also needs to be implemented in the daily lives of parents and children. However, the present research is limited by the use an old version of the consulted database, which does not describe species of bacteria.

## Figures and Tables

**Figure 1 microorganisms-10-01748-f001:**
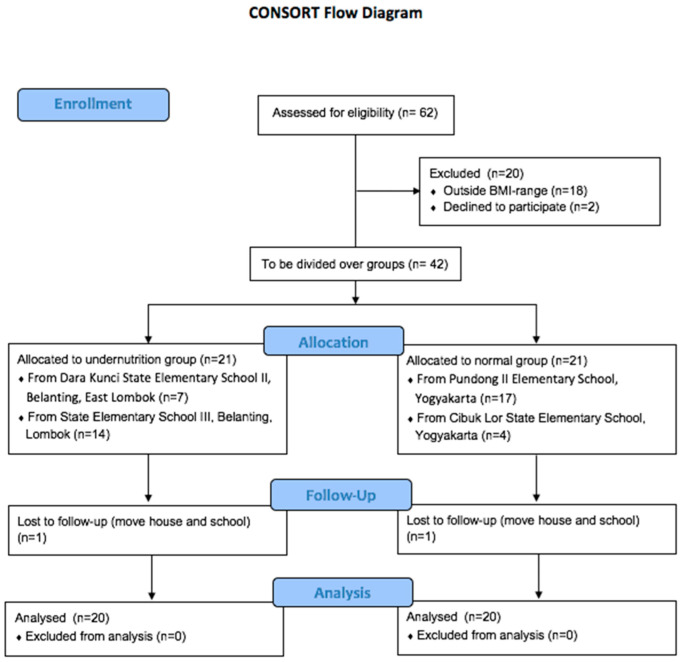
CONSORT flow diagram of the case–control study.

**Figure 2 microorganisms-10-01748-f002:**
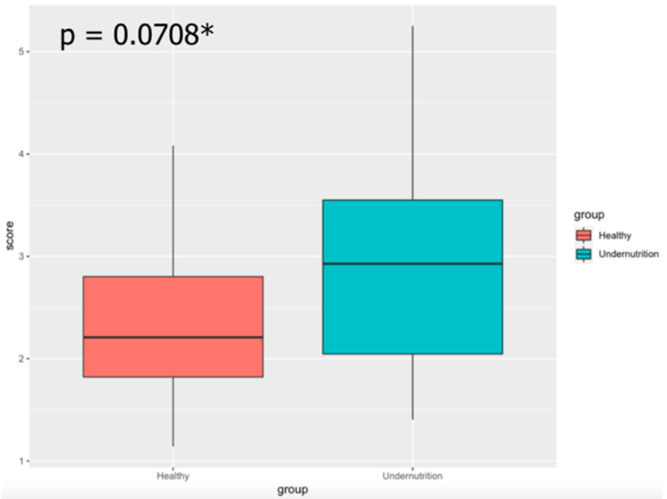
Ratio of Firmicutes to Bacteroidetes. Significant differences between groups were calculated using the Mann–Whitney U test (* *p* < 0.1).

**Figure 3 microorganisms-10-01748-f003:**
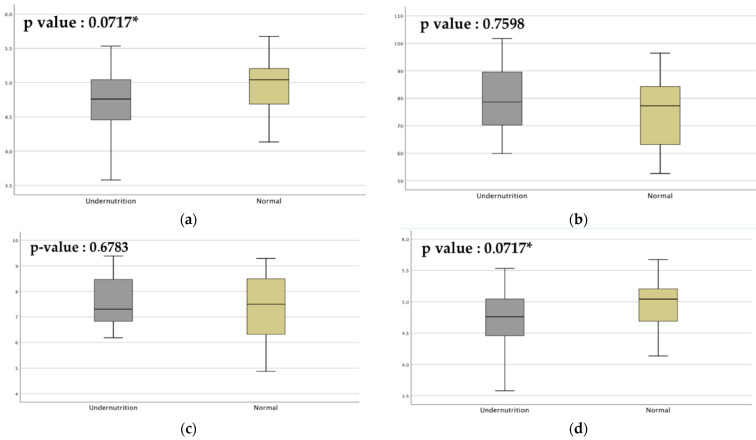
The significance of gut microbiota alpha diversity was calculated using an independent *t*-test (* *p* < 0.1). (**a**) Chao1, species abundance between groups; (**b**) observed OTUs, number of species or OTUs based on the community evenness; (**c**) whole-tree PD, comparison of phylogenetic diversity; (**d**) Shannon index, species diversity between groups.

**Figure 4 microorganisms-10-01748-f004:**
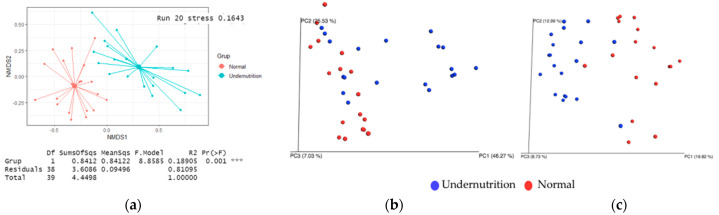
Beta diversity of gut microbiota composition. (**a**) Bray–Curtis; (**b**) weighted UniFrac; (**c**) Unweighted UniFrac.

**Figure 5 microorganisms-10-01748-f005:**
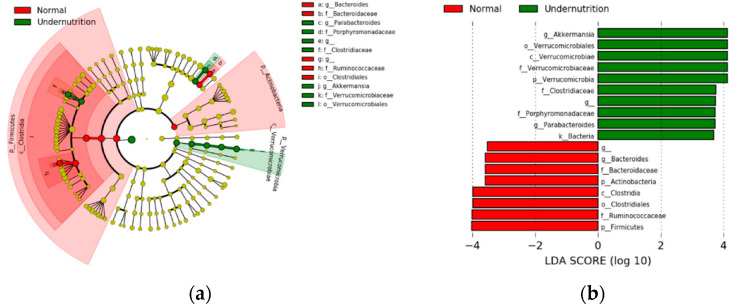
Biomarker determination of gut microbiota using the taxonomic profile method of LefSe (**a**) and linear discriminant analysis score (**b**). The bar chart and cladogram show taxonomic discrimination between the two groups based on the cladogram and LDA method. Taxa with an alpha value of 0.05 and LDA score < 3.5 were considered significant.

**Figure 6 microorganisms-10-01748-f006:**
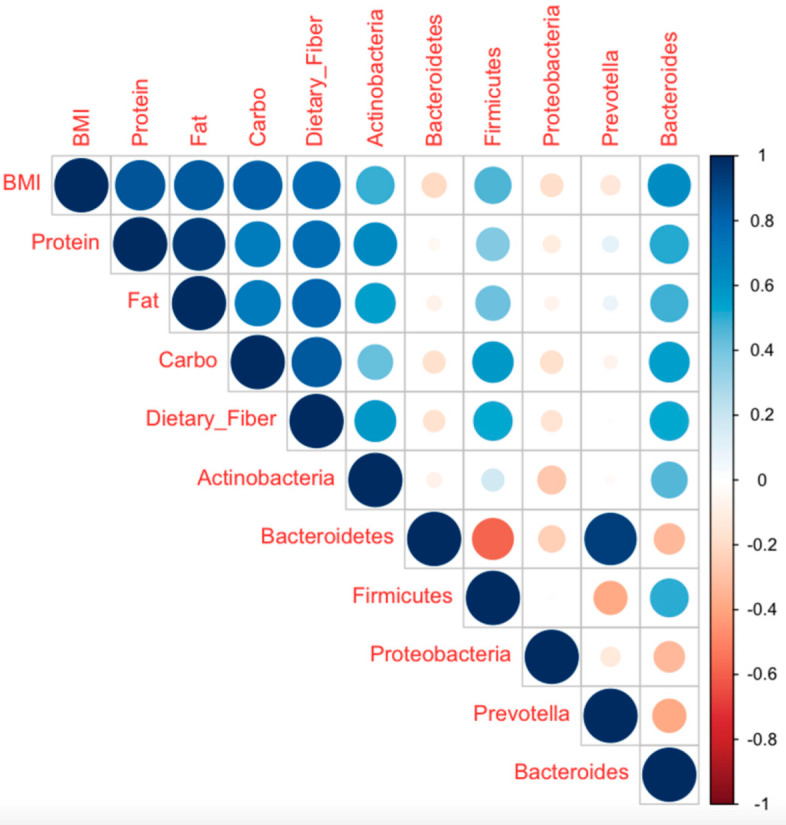
Correlation between diet and gut microbiota composition generated using corrplot. The blue color indicates a positive correlation between two parameters, whereas red indicates that the two parameters are negatively correlated.

**Figure 7 microorganisms-10-01748-f007:**
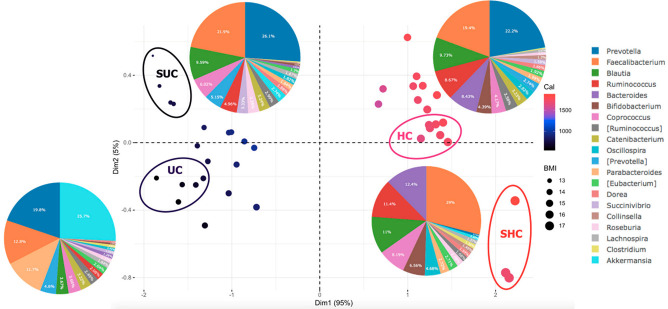
Principal component analysis (PCA) plot of 40 children related to calorie intake and BMI. The calorie intake (Cal) and BMI of each sample are represented by dots and color, respectively. The group sample was selected based on the four regions and circled with letters denoting the groups, namely SUB (severely undernourished children), UC (undernourished children), HC (healthy children), and SHC (super healthy children). The genus composition was averaged and graphed in the pie charts.

**Table 1 microorganisms-10-01748-t001:** Characteristics of 40 Indonesian subjects in this study.

Category	Normal (n = 20)	Undernutrition (n = 20)	*p*-Value ^1^
Male	6 (30%)	12 (60%)	
Female	14 (70%)	8 (40%)	
Age (years)	9.65 ± 0.67	9.99 ± 0.99	0.063
Body weight (kg)	27.82 ± 3.21	21.57 ± 2.61	<0.001
Body height (cm)	131.69 ± 5.54	129.31 ± 6.38	0.123
BMI (kg/m^2^)	15.98 ± 0.73	12.87 ± 0.4	0.001

^1^ Data are presented as means ± SD. Significant differences between the groups were calculated using the Wilcoxon rank sum test (*p* < 0.05).

**Table 2 microorganisms-10-01748-t002:** Dietary intake of normal vs. undernourished children.

Nutrient Intake	Normal	Undernourished	RDA	%RDA		*p*-Value ^1^
				Normal	Undernourished	
Energy (kcal)	1718.3 ± 74.67	779 ± 122.81	2036.3	84.38 ± 3.67	38.26 ± 6.03	<0.001
Macronutrient						
Protein (g)	72.64 ± 10.65	25.69 ± 6.21	60.1	17.3 ± 2.29	13.4 ± 2.76	<0.001
Fat (g)	80.57 ± 6.8	23.96 ± 5.99	69.1	41.65 ± 3.79	27.4 ± 5.92	<0.001
Carbohydrate (g)	173.64 ± 21.9	114.34 ± 23.98	290.7	41.15 ± 5.19	59.2 ± 7.23	<0.001
Dietary fiber (g)	7.51 ± 1.2	3.59 ± 1.49	28	26.81 ± 4.29	12.85 ± 5.31	<0.001
PUFA (g)	16.15 ± 2.9	5.79 ± 3.33	10	16.15 ± 2.9	57.86 ± 33.26	<0.001
Vitamin						
Vitamin A (μg)	1089.84 ± 429.04	279.96 ± 129.92	900	121.09 ± 47.67	31.11 ± 14.44	<0.001
Vitamin E (mg)	4.22 ± 1.13	2.07 ± 0.96	11	38.4 ± 10.30	18.80 ± 8.74	<0.001
Vitamin B1 (mg)	0.53 ± 0.09	0.23 ± 0.08	1	53.5 ± 5.19	23.2 ± 7.75	<0.001
Vitamin B2 (mg)	1.03 ± 0.25	0.35 ± 0.13	1.2	86.13 ± 21.05	28.79 ± 10.8	<0.001
Vitamin B6 (mg)	1.05 ± 0.15	0.45 ± 0.13	1	105.15 ± 14.63	45 ± 13.07	<0.001
Vitamin C (mg)	19.29 ± 9.54	21.09 ± 19.43	90	21.44 ± 10.60	23.44 ± 21.59	0.725
Folic acid (μg)	116.06 ± 24.51	63.45 ± 34.54	400	29.02 ± 6.13	15.86 ± 8.64	<0.001
Mineral						
Sodium (mg)	761.49 ± 269.16	337.98 ± 470.85	2000	38.07 ± 13.46	16.90 ± 23.54	<0.001
Potassium (mg)	1227.32 ± 234.58	530.59 ± 158.89	2000	61.37 ± 11.73	26.53 ± 7.94	<0.001
Calcium (mg)	334.45 ± 140.37	211.94 ± 100.56	1100	30.4 ± 12.76	19.27 ± 9.14	0.001
Magnesium (mg)	175.65 ± 23.39	102.73 ± 32.44	250	70.26 ± 9.38	41.09 ± 12.97	<0.001
Phosphorus (mg)	844.21 ± 127.38	358.76 ± 90.99	1250	67.54 ± 10.19	28.7 ± 7.28	<0.001
Iron (mg)	7.31 ± 2.35	3.26 ± 0.96	15	48.73 ± 15.66	21.7 ± 6.37	<0.001
Zinc (mg)	8.04 ± 1.29	3.56 ± 0.65	7	114.82 ± 18.47	50.87 ± 9.35	<0.001

^1^ Data are presented as means ± SD. A significant difference between the two groups was calculated using the Wilcoxon rank sum test (*p* < 0.05).

**Table 3 microorganisms-10-01748-t003:** Phylum level of gut microbiota composition of 40 Indonesian subjects.

Phylum	Relative Abundance		*p* Value ^1^
	Normal	Undernourished	
Firmicutes	65.76 ± 9.94	62.18 ± 12.03	0.044
Bacteroidetes	29.97 ± 8.05	23.47 ± 7.89	0.014
Actinobacteria	4.84 ± 4.30	0.85 ± 0.32	<0.001
Proteobacteria	1.96 ± 4.29	2.56 ± 2.49	<0.001
Verrucomicrobia	0.18 ± 0.41	13.46 ± 12.52	0.007
Fusobacteria	0.12 ± 0.38	0.10 ± 0.29	0.779

^1^ Data are shown as means ± standard deviation (%). Significant differences between groups were calculated using the Wilcoxon rank sum test (*p* < 0.05).

**Table 4 microorganisms-10-01748-t004:** Genus-level gut microbiota composition of 40 Indonesian subjects.

Phylum	Genus	Relative Abundance		*p*-Value ^1^
		Normal	Undernourished	
Firmicutes	*Caprococus*	3.83 ± 1.83	3.23 ± 1.83	0.265
	*Ruminococcus*	5.09 ± 3.14	2.28 ± 1.50	0.002
	*Roseburia*	1.01 ± 0.85	1.52 ± 1.61	0.201
	*Clostridium*	0.28 ± 0.33	0.34 ± 0.43	0.904
	*Blautia*	5.67 ± 1.95	4.77 ± 3.01	0.142
	*Faecalibacterium*	12.65 ± 4.41	9.60 ± 5.00	0.086
	*Lachnospira*	0.63 ± 0.47	0.92 ± 0.97	0.583
Bacteroidetes	*Prevotella*	15.36 ± 14.28	13.45 ± 9.34	0.841
	*Bacteroides*	4.30 ± 3.93	0.72 ± 0.38	<0.001
Actinobacteria	*Bifidobacterium*	3.26 ± 2.9	0.35 ± 0.30	<0.001
	*Collinsella*	0.82 ± 0.43	0.23 ± 0.13	<0.001
Proteobacteria	*Succinivibrio*	1.60 ± 3.94	1.93 ± 2.38	0.004
Verrucomicrobia	*Akkermansia*	0.15 ± 0.37	12.56 ± 12.35	0.002
Fusobacteria	*Fusobacterium*	0.10 ± 0.35	0.09 ± 0.27	0.883

^1^ Data are shown as means ± standard deviation (%). Significant differences between groups were calculated with the Wilcoxon rank sum test (*p* < 0.05).

**Table 5 microorganisms-10-01748-t005:** Correlation between sociodemographic factors and gut microbiota composition in undernourished children.

	Gender	BMI	Age	Parents‘ Education	Income	Medical Record	Sanitation	Parents‘ Occupation	Parents‘ Knowledge	Exclusive Breastfeeding	Actinobacteria	Bacteroidetes	Firmicutes	Proteobacteria	*Prevotella*
BMI	0.34														
Age	0.30	0.11													
Parents‘ education	−0.26	0.09 ^a^	−0.25												
Income	0.22	0.36	−0.04 ^a^	0.41											
Medical record	0.08 ^a^	−0.02 ^a^	−0.05 ^a^	0.23	0.17										
Sanitation	−0.15	0.13	−0.40	0.48	0.32	0.21									
Parents‘ occupation	0.13	0.34	−0.07	0.30	0.76	0.16	0.33								
Parents‘ knowledge	0.11	0.01 ^a^	0.16	0.45	0.30	0.09 ^a^	0.42	0.20							
Exclusive breastfeeding	−0.18	0.08 ^a^	−0.01 ^a^	−0.08 ^a^	−0.04 ^a^	−0.04 ^a^	−0.04 ^a^	0.01 ^a^	−0.12						
Actinobacteria	−0.21	0.50	−0.20	0.33	0.13	0.12	0.39	0.26	0.33	−0.25					
Bacteroidetes	0.54	−0.21	−0.13	0.07	−0.10	0.13	−0.02 ^a^	−0.15	−0.02 ^a^	−0.14	−0.08 ^a^				
Firmicutes	−0.44	0.46	0.02 ^a^	0.26	0.15	0.02 ^a^	0.37	0.26	0.17	0.29	0.17	−0.59			
Proteobacteria	−0.06 ^a^	−0.18	0.21	−0.13	−0.07 ^a^	0.18	−0.17	0.12	−0.06	−0.22	−0.27	−0.24	−0.02 ^a^		
*Prevotella*	0.43	−0.13	0.18	0.12	−0.11	0.20	0.08 ^a^	−0.11	0.00 ^a^	−0.11	−0.03 ^a^	0.93	−0.38	−0.13	
*Bacteroides*	−0.38	0.62	−0.06 ^a^	0.41	0.16	−0.15	0.43	0.16	0.35	−0.07 ^a^	0.45	−0.32	−0.5	−0.33	−0.39

^a^ Significant differences were calculated using the Pearson correlation coefficient of all variables (*p* < 0.1). The sign in front of coefficients describes the correlation (negatively correlated if the sign is negative).

## Data Availability

The data presented in this study are available upon request from the corresponding author.

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
