# Peer review of "Gut Microbiota Composition in Undernourished Children Associated with Diet and Sociodemographic Factors: A Case–Control Study in Indonesia"

_microorganisms, 2022, doi:10.3390/microorganisms10091748_

Round 1
Reviewer 1 Report
Gatya et al., manuscript with the title Gut Microbiota Composition in Undernourished Children As-2 sociated with Diet and Sociodemographic Factors: A Case-Con-trol Study in Indonesia is presenting difference in microbiota composition in the undernourished children. They analyzed composition of the microbiota with 16S r gene amplification sequencing. With this technique they analyzed the composition of microbiota in different groups of undernourished children. Additionally, they analyzed the composition of the diet in undernourished and normal nourished group of children,
The reviewer thinks the topic of the paper is interesting and will be beneficial for development of new diagnostic methods of the treatment of children disease relate to intestine.
In the paper there are many diet sources that are decreased in children that are undernourished. Interesting the level of vitamin C remain at the similar level in normal or undernourished children. Do you have any possible explanation additional to the fruits they are eating such as banana and papaya for these? Did you analyzed any additionally vitamins and proteins that are present in these two fruits?
Figure 6 and table 4 are confusing comparison for me. In figure 6 you are compering impact of different diets on the composition of microbiota and the same time you are also comparing diet to diet and different strains of microorganism between each other? That to me seems unfeasible to do or you would need more precisely define the groups you are comparing. Similar in the table 4 you are comparing all the different impact on the diet and also comparing diets between each other.
In the figure 6 you are analyzing different groups of diets (SUC, UC, SHC,HC). In the manuscript you are not explain what is the exact diet of each group. That needs to be added. You would also need what is the outcome/difference between these 4 diets.
The message of the paper needs to be set more exact so that the people will see the benefit for the diagnostic in microbiota and composition of diets in the treatments of disease related to different diets. It is comparing to many different aspects and the outcome is to dispersed. It had to have a more simple and exact message.
Reviewer 2 Report
The manuscript is sound and well written. I suggest authors include the catalog number of chemicals purchased in the materials section.
